# Pb-Hash: Partitioned b-bit Hashing

## ABSTRACT

Many hashing algorithms including minwise hashing (MinHash), one permutation hashing (OPH), and consistent weighted sampling (CWS) generate integers of $B$ bits. With $k$ hashes for each data vector, the storage would be $B \times k$ bits; and when used for large-scale learning, the model size would be $2^B \times k$, which can be expensive. A standard strategy is to use only the lowest $b$ bits out of the $B$ bits and somewhat increase $k$, the number of hashes. In this study, we propose to re-use the hashes by partitioning the $B$ bits into $m$ chunks, e.g., $b \times m = B$. Correspondingly, the model size becomes $m \times 2^b \times k$, which can be substantially smaller than $2^B \times k$.

There are multiple reasons why the proposed "partitioned b-bit hashing" (Pb-Hash) can be desirable: (1) Generating hashes can be expensive for industrial-scale systems especially for many user-facing applications. Thus, engineers may hope to make use of each hash as much as possible, instead of generating more hashes (i.e., by increasing the $k$). (2) To protect user privacy, the hashes might be artificially "polluted" and the differential privacy (DP) budget is proportional to $k$. (3) After hashing, the original data are not necessarily stored and hence it might not be even possible to generate more hashes. (4) One special scenario is that we can also apply Pb-Hash to the original categorical (ID) features, not just hashed data.

Our theoretical analysis reveals that by partitioning the hash values into $m$ chunks, the accuracy would drop. In other words, using $m$ chunks of $B/m$ bits would not be as accurate as directly using $B$ bits. This is due to the correlation from re-using the same hash. On the other hand, our analysis also shows that the accuracy would not drop much for (e.g.,) $m = 2 \sim 4$. In some regions, Pb-Hash still works well even for $m$ much larger than 4. We expect Pb-Hash would be a good addition to the family of hashing methods/applications and benefit industrial practitioners.

We verify the effectiveness of Pb-Hash in machine learning tasks, for linear SVM models as well as deep learning models. Since the hashed data are essentially categorical (ID) features, we follow the standard practice of using embedding tables for each hash. With Pb-Hash, we need to design an effective strategy to combine $m$ embeddings. Our study provides an empirical evaluation on four pooling schemes: concatenation, max pooling, mean pooling, and product pooling. There is no definite answer which pooling would be always better and we leave that for future study.

*ICTIR '24, Washington D.C.,*
© 2018 Association for Computing Machinery.
ACM ISBN 978-x-xxxx-xxxx-x/YY/MM...$15.00
https://doi.org/XXXXXXX.XXXXXXX

**ACM Reference Format:**
Anonymous Author(s). 2018. Pb-Hash: Partitioned b-bit Hashing. In *Proceedings of July 14–18 (ICTIR '24)*. ACM, New York, NY, USA, 9 pages. https://doi.org/XXXXXXX.XXXXXXX

## 1 INTRODUCTION

In this paper, we focus on effectively re-using hashes and developing the theory to explain some of the interesting empirical observations. Typically, for each data vector, applying some hashing method $k$ times generates $k$ integers of $B$ bits, where $B$ can be (very) large. For example, with the celebrated minwise hashing [4–6, 33, 34], we generate a permutation of length $D$, where $D$ is the data dimension, and apply the same permutation to all data vectors (which are assumed to be binary). For each data vector, the location of the first non-zero entry after the permutation is the hashed value. Then we repeat the permutation process $k$ times to generate $k$ hash values for each data vector. For vector $u$, we denote its $k$ hashes as $h_j(u)$, $j = 1, 2, ..., k$. For vector $v$, we similarly have $h_j(v)$. It is known that the collision probability is $Pr(h_j(u) = h_j(v)) = J$, where for minwise hashing $J$ is the Jaccard similarity between two binary vectors $u$ and $v$, i.e., $J = \frac{\sum_{i=1}^{D} 1\{u_i \neq 0 \text{ and } v_i \neq 0\}}{\sum_{i=1}^{D} 1\{u_i \neq 0 \text{ or } v_i \neq 0\}}$.

When we use (e.g.,) minwise hashes for building machine learning models, we need to treat the hash values as categorical features and expand them as one-hot representations. For example, if $D = 4$, then the minwise hash values are between 0 and 3. Supposed $k = 3$ hashes are {3, 1, 2}, we will encode them as a $2^2 \times 3 = 12$-dimensional binary vector: $[1, 0, 0, 0, \ 0, 0, 1, 0, \ 0, 1, 0, 0]$ as the feature vector fed to the model. Let $D = 2^B$. This scheme can easily generate extremely high-dimensional data vectors and excessively large model sizes. A common strategy is to only use the lowest $b$ bits for each hash value, a method called "b-bit minwise hashing" [34]. It can be a drastic reduction from $2^B$ is $2^b$, for example, $B = 32$ and $b = 10$. Typically, we will have to increase $k$ the number of hashes to compensate the loss of accuracy due to the use of only $b$ bits.

### 1.1 Collision Probability of $b$-bit Hashing and the Basic Assumption

Denote $h_j^{(b)}(u)$ and $h_j^{(b)}(v)$ as the lowest $b$ bits of $h_j(u)$ and $h_j(v)$, respectively. Theorem 1.1 describes the collision probability of minwise hashing $Pr\left(h_j^{(b)}(u) = h_j^{(b)}(v)\right)$ by assuming $D = 2^B$ is large.

THEOREM 1.1. [34] $Pr\left(h_j(u) = h_j(v)\right) = J$ is the collision probability of minwise hashing. Assume $D$ is large. Denote $f_1 = \sum_{i=1}^{D} 1\{u_i \neq 0\}$, $f_2 = \sum_{i=1}^{D} 1\{v_i \neq 0\}$. Then

$$P_b = Pr\left(h_j^{(b)}(u) = h_j^{(b)}(v)\right) = C_{1,b} + (1 - C_{2,b})J \qquad (1)$$

*where*

$$C_{1,b} = A_{1,b}\frac{r_2}{r_1+r_2} + A_{2,b}\frac{r_1}{r_1+r_2}, \qquad C_{2,b} = A_{1,b}\frac{r_1}{r_1+r_2} + A_{2,b}\frac{r_2}{r_1+r_2},$$

$$A_{1,b} = \frac{r_1[1-r_1]^{2^b-1}}{1-[1-r_1]^{2^b}}, \qquad A_{2,b} = \frac{r_2[1-r_2]^{2^b-1}}{1-[1-r_2]^{2^b}},$$

$$r_1 = \frac{f_1}{D}, \qquad r_2 = \frac{f_2}{D}$$

The result in Theorem 1.1 was obtained via conducting careful and tedious summations of the individual probability terms. Interestingly, if $r_1, r_2 \to 0$, then $A_{1,b} = A_{2,b} = \lim_{r\to 0}\frac{r[1-r]^{2^b-1}}{1-[1-r2^b]} = \frac{1}{2^b}$, $C_{1,b} = C_{2,b} = \frac{1}{2^b}$ and $P_b = \frac{1}{2^b} + \left(1 - \frac{1}{2^b}\right)J = J + (1-J)\frac{1}{2^b}$. This (much) simplified probability has an intuitive interpretation using (approximate) conditional probabilities: $h_j(u) = h_j(v)$ with probability $J$. If $h_j(u) \neq h_j(v)$ (which occurs with probability $(1-J)$), there is still a roughly $\frac{1}{2^b}$ probability to have $h_j^{(b)}(u) = h_j^{(b)}(v)$, because the space is of size $2^b$. In fact, one can also resort to the commonly used "re-hash" idea to explicitly map $h_j(u)$ uniformly into $[0, 1, 2, ..., 2^b - 1]$.

Therefore, in this paper, we make the following basic assumption:

**Basic Assumption:** Apply the hash function $h$ to two data vectors $u$ and $v$ to obtain $h(u)$ and $h(v)$, respectively, where $h(.) \in [0, 1, 2, ..., 2^B - 1]$. The collision probability is $Pr(h(u) = h(v)) = J$. $h^{(b)}(u)$ and $h^{(b)}(v)$ denote the values by taking $b$ bits of $h(u)$ and $h(v)$, respectively, with

$$P_b = Pr\left(h^{(b)}(u) = h^{(b)}(v)\right) = c_b + (1-c_b)J, \qquad c_b = \frac{1}{2^b} \quad (2)$$

We call it an "assumption" because, when the original space is large, the "re-hash" trick typically can only be done approximately, for example, through universal hashing [8]. There is also an obvious "descrepancy" that, in (2), we actually need $b \to \infty$ in order to have $Pr\left(h^{(b)}(u) = h^{(b)}(v)\right) = J$. But here for simplicity we just assume that, when $b = B$, we have $Pr\left(h^{(B)}(u) = h^{(B)}(v)\right) = J$. Because $B$ is typically large, we do not worry much about the discrepancy. Otherwise the analysis would be too complicated, just like Theorem 1.1.

The basic assumption (2) allows us to derive a simple unbiased estimator of the basic similarity $J$:

$$\hat{J}_b = \frac{\hat{P}_b - c_b}{1 - c_b}, \qquad Var\left(\hat{J}_b\right) = \frac{Var(\hat{P}_b)}{(1-c_b)^2} = \frac{P_b(1-P_b)}{(1-c_b)^2}. \quad (3)$$

where the variance $Var\left(\hat{J}_b\right)$ assumes only one sample, because the sample size $k$ will usually be canceled out in the comparison. When $b = B$, the variance of $\hat{J}$ would be simply $J(1-J)$, i.e., the variance of the Bernoulli trial. We can compute the ratio of the variances to assess the loss of accuracy due to taking only $b$ bits:

$$R_b = \frac{Var(\hat{J}_b)}{Var(\hat{J})} = \frac{P_b(1-P_b)}{(1-c_b)^2}\frac{1}{J(1-J)} \quad (4)$$

$$= 1 + \frac{c_b}{1-c_b}\frac{1}{J} = 1 + \frac{1}{(2^b-1)J}$$

Here $R_b$ (where $R_b \to \infty$ as $J \to 0$) can be viewed as the multiplier needed for increasing the sample size by using only $b$ bits. In real-world applications, typically only a tiny fraction of data vector pairs have relatively large similarity ($J$) values. For the majority of the pairs, the $J$ values are very small. For example, when $J = 0.1$ and $b = 1$, we have $R_b = 11$. In other words, if we keep only 1 bit per hash and increase the number of hashes by a factor of 11, then the variance would remain the same.

## 1.2 Motivations for Re-using Hashes and Pb-Hash: Partitioned b-bit Hashing

Instead of using fewer bits and generating more hashes, in this paper, we study the strategy of re-using the hashes. The idea is simple. For a $B$-bit hash value, we break the bits into $m$ chunks: $b_1, b_2, ..., b_m$ with $\sum_{i=1}^{m} b_i = B$. It is often convenient to simply let $b_1 = b_2 = ... = b_m = b$ and $m \times b = B$. The dimensionality is (substantially) reduced from $2^B$ to $m \times 2^b$. In many scenarios, this strategy can be desirable. In industrial large-scale systems, the cost for generating hashes can often be considerable especially for serving (for example, in many user-facing applications). Thus, it is always desired if we can generate fewer hashes for better efficiency. From the perspective of privacy protection, it is also crucial to reduce $k$ the number of hashes, because typically the needed privacy budget "$\epsilon$" (in the $(\epsilon, \delta)$-DP language [15]) is proportional to $k$. There is another strong motivation in that we may not be able to generate more hashes in some situations. For example, in some applications, the original data are not necessarily stored after hashing.

Interestingly, we can also directly apply the Pb-Hash idea to the original categorical (ID) features. In large-scale recommender systems [17, 51, 60], the use of ID features is dominating. For companies which do not have infrastructure to handle ID features of billion or even just million categories, they can apply Ph-Hash to reduce the model dimensions.

Figure 1 is an illustration of the idea of Pb-Hash with training for large ID data. Basically, we can first apply a random permutation on the IDs, then break the bits into $m$ chunks so that one can substantially reduce the embedding size, for example, from the original size of $2^B$ to $m \times 2^b$ with $B = m \times b$. The number of parameters will be substantially reduced. We will need a strategy to merge these $m$ embedding tables. The obvious choices are concatenation, mean, max, and product. Note that for this application, our Pb-Hash includes the so-called "QR-hash" [51] as a special case (which uses $m = 2$).

## 2 THEORETICAL ANALYSIS OF PB-HASH

Recall the **Basic Assumption**: $P_b = Pr\left(h^{(b)}(u) = h^{(b)}(v)\right) = c_b + (1-c_b)J$, $c_b = \frac{1}{2^b}$. With Pb-Hash, the basic idea is to break the total $B$ bits into $m$ chunks. Let $\sum_{i=1}^{m} b_i = B$, and later we can assume $b_1 = b_2 = ... = b_m$ to simplify the expressions. Then, we have the following expectations:

$$E\left(\hat{P}_{b_i}\right) = c_{b_i} + (1-c_{b_i})J \quad (5)$$

$$E\left(\sum_{i=1}^{m} \hat{P}_{b_i}\right) = \sum_{i=1}^{m} c_{b_i} + J\sum_{i=1}^{m}(1-c_{b_i}). \quad (6)$$

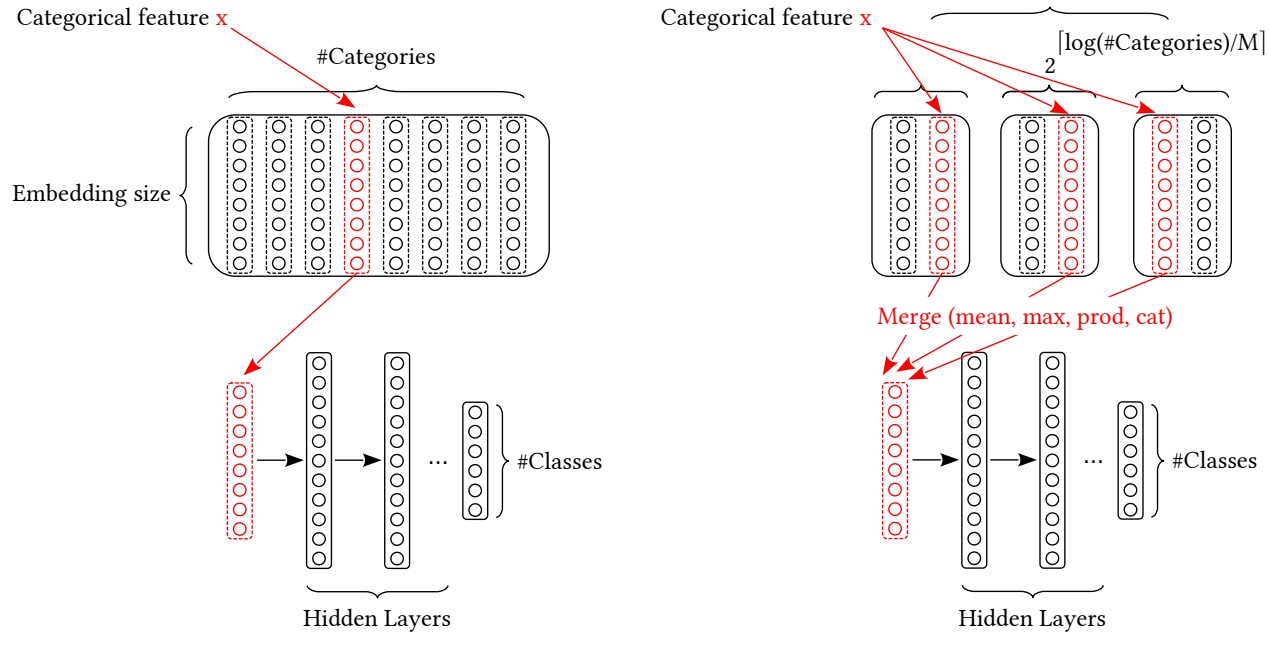

**Embedding table lookup**                    **Pb-Hash lookup**

Figure 1: An visual illustration for the embedding table lookup and Pb-Hash lookup.

which allows us to write down an unbiased estimator of $J$:

$$\hat{J}_m = \frac{\sum_{i=1}^m \hat{P}_{b_i}}{\sum_{i=1}^m (1 - c_{b_i})} - \frac{\sum_{i=1}^m c_{b_i}}{\sum_{i=1}^m (1 - c_{b_i})}. \tag{7}$$

THEOREM 2.1.

$$E\left(\hat{J}_m\right) = J, \tag{8}$$

$$Var\left(\hat{J}_m\right) = \frac{\sum_{i=1}^m P_{b_i}(1 - P_{b_i}) + \sum_{i\neq i'} \left(P_{b_i+b_{i'}} - P_{b_i}P_{b_{i'}}\right)}{\left(\sum_{i=1}^m (1 - c_{b_i})\right)^2}. \tag{9}$$

where $\quad c_{b_i} = \dfrac{1}{2^{b_i}}, \quad P_{b_i} = c_{b_i} + (1 - c_{b_i})J, \tag{10}$

$$P_{b_i+b_{i'}} = c_{b_i+b_{i'}} + (1 - c_{b_i+b_{i'}})J \tag{11}$$

**Proof of Theorem 2.1.** Firstly, it is easy to show that

$$E\left(\hat{J}_m\right) = J, \qquad Var\left(\hat{J}_m\right) = Var\left(\sum_{i=1}^m \hat{P}_{b_i}\right) \bigg/ \left(\sum_{i=1}^m (1 - c_{b_i})\right)^2.$$

Then we expand the variance of the sum:

$$Var\left(\sum_{i=1}^m \hat{P}_{b_i}\right) = \sum_{i=1}^m Var\left(\hat{P}_{b_i}\right) + \sum_{i\neq i'} Cov\left(\hat{P}_{b_i}, \hat{P}_{b_{i'}}\right)$$

$$= \sum_{i=1}^m P_{b_i}(1 - P_{b_i}) + \sum_{i\neq i'} \left(P_{b_i+b_{i'}} - P_{b_i}P_{b_{i'}}\right).$$

Here we have used the **Basic Assumption.** □

The key in the analysis is the covariance term $Cov\left(\hat{P}_{b_i}, \hat{P}_{b_{i'}}\right)$, which in the independence case would be just zero. With Pb-Hash, however, the covariance is always non-negative. This is the reason why the accuracy of using $m$ chunks of $b$-bits from the same hash value would not be as good as using $m$ independent $b$-bits (i.e., $m$ independent hashes).

LEMMA 2.2.

$$P_{b_1+b_2} - P_{b_1}P_{b_2} \geq 0 \tag{12}$$

is a concave function in $J \in [0, 1]$. Its maximum is $\frac{1}{4}\left(1 - \frac{1}{2^{b_1}}\right)\left(1 - \frac{1}{2^{b_2}}\right)$, attained at $J = 1/2$.

**Proof of Lemma 2.2**

$$f(J) = P_{b_1+b_2} - P_{b_1}P_{b_2}$$

$$= J + (1 - J)\frac{1}{2^{b_1+b_2}} - \left(J + (1 - J)\frac{1}{2^{b_1}}\right)\left(J + (1 - J)\frac{1}{2^{b_2}}\right)$$

$$f''(J) = -\left(1 - \frac{1}{2^{b_1}}\right)\left(1 - \frac{1}{2^{b_2}}\right) \leq 0$$

This means that $f(J)$ is a concave function in $J \in [0, 1]$. Also, we have

$$f(0) = \frac{1}{2^{b_1+b_2}} - \frac{1}{2^{b_1}}\frac{1}{2^{b_2}} = 0, \qquad f(1) = 1 - 1 = 0$$

Therefore, we must have $f(J) \geq 0$. Furthermore, by setting $f'(J) = 0$, we can see that the maximum value of $f(J)$ is attained at $J = 1/2$. □

Figure 2 verifies the results in Lemma 2.2, with $P_{2b} - P_b^2$ (left panel) and $P_{2b} - P_1 P_{2b-1}$ (right panel). It is interesting that in both cases, the maximums are attained at $J = 1/2$, as predicted.

To simplify the expression and better visualize the results, we consider $b_1 = b_2 = ... = b_m = b$ and $b \times m = B$. Then we have

$$\hat{J}_m = \frac{\sum_{i=1}^m \hat{P}_{b_i}}{m(1 - c_b)} - \frac{c_b}{1 - c_b}, \tag{13}$$

and

$$Var\left(\hat{J}_m\right) = \frac{P_b(1 - P_b) + (m - 1)\left(P_{2b} - P_b^2\right)}{m(1 - c_b)^2} \tag{14}$$

$$= \frac{1}{m} \frac{P_b(1 - P_b)}{(1 - c_b)^2} + \frac{m - 1}{m} \frac{P_{2b} - P_b^2}{(1 - c_b)^2}.$$

We can again compare the variance of $Var\left(\hat{J}_m\right)$ with, $J(1 - J)$, which is the variance of $\hat{J}$ using all the bits:

$$R_{m,b} = \frac{Var\left(\hat{J}_m\right)}{J(1 - J)} = \frac{P_b(1 - P_b) + (m - 1)\left(P_{2b} - P_b^2\right)}{m(1 - c_b)^2 J(1 - J)}, \quad m \times b = B. \tag{15}$$

When $R_{m,b}$ is close to 1, it means that Pb-Hash does not lose accuracy as much. Recall that, if we have hashed values for building learning models, the model size is $2^B \times k$, where $k$ is the number of hashes. By Pb-Hash, we can (substantially) reduce the model size to be $m \times 2^b \times k$. In practice, the ID features can have very high cardinality, for example, a million (i.e., $B = 20$) or billion (i.e., $B = 30$). Figure 3 implies that, as long as $B$ is not too small, we do not expect a significant loss of accuracy if $m = 2 \sim 4$.

## 3 APPLICATIONS AND EXPERIMENTS

Recall that in our **Basic Assumption**, we have not specified which particular hashing method is used. For the applications and experiments, we focus on minwise hashing (MinHash) for binary (0/1) data, and consistent weighted sampling (CWS) for general non-negative data.

### 3.1 Minwise Hashing (MinHash) on Binary Data

The binary Jaccard similarity, also known as the "resemblance", is a similarity metric widely used in machine learning and web applications. It is defined for two binary (0/1) data vectors, denoted as $u$ and $v$, where each vector belongs to the set $\{0, 1\}^D$. The Jaccard similarity is calculated as:

$$J(u, v) = \frac{\sum_{i=1}^D 1\{u_i = v_i = 1\}}{\sum_{i=1}^D 1\{u_i + v_i \geq 1\}}. \tag{16}$$

In this context, the vectors $u$ and $v$ can be interpreted as sets of items, represented by the positions of non-zero entries. However, computing pairwise Jaccard similarity becomes computationally expensive as the data size increases in industrial applications with massive datasets. To address this challenge and enable large-scale search and learning, the "minwise hashing" (MinHash) algorithm is introduced [4–6, 33, 34] as a standard hashing technique for approximating the Jaccard similarity in massive binary datasets.

MinHash has found applications in various domains, including near neighbor search, duplicate detection, malware detection, clustering, large-scale learning, social networks, and computer vision [2, 7, 9–12, 14, 18, 19, 24, 26, 28, 30, 44, 45, 53, 54, 57, 62].

MinHash produces integer outputs. For efficient storage and utilization of the hash values in large-scale applications, Li and König [34] proposed a variant called "$b$-bit MinHash". This method only retains the lowest $b$ bits of the hashed integers, providing a memory-efficient and convenient approach for similarity search and machine learning tasks. Over the years, $b$-bit MinHash has become the standard implementation of MinHash [31, 35, 50, 59]. Additionally, we should mention "circulant MinHash" (C-MinHash) [40]. C-MinHash employs a single circular permutation, which enhances hashing efficiency and perhaps surprisingly improves estimation accuracy. Figure 4 depicts the use case of Pb-Hash on minwise hashing, for verifying the theoretical results in Theorem 2.1.

### 3.2 Consistent Weighted Sampling (CWS) and Linear SVM

MinHash and OPH are techniques designed to process binary data, representing unweighted sets. In the literature, to tackle the real-valued data, the weighted Jaccard similarity is defined as follows:

$$J(u, v) = \frac{\sum_{i=1}^D \min\{u_i, v_i\}}{\sum_{i=1}^D \max\{u_i, v_i\}},$$

where $u, v \in \mathbb{R}_+^D$ are two non-negative data vectors. In contrast to binary data, weighted data often carries more detailed information. Consequently, the weighted Jaccard similarity measure has garnered significant attention and has been extensively studied and applied across various domains, such as theory, databases, machine learning, and information retrieval [1, 3, 9, 13, 19–21, 23, 27, 29, 38, 43, 46–49, 55, 56, 58, 61, 63]. This extended similarity metric enables the analysis and comparison of weighted sets, facilitating a deeper understanding of the underlying data. ChatGPT The weighted Jaccard similarity has emerged as a potential non-linear kernel, especially in the realm of large-scale classification and regression tasks [32]. It has been demonstrated to surpass the widely used RBF (Radial Basis Function) kernel in terms of performance across numerous tasks and datasets. The weighted Jaccard similarity's ability to capture intricate relationships within the datasets it is applied to makes it a promising choice for achieving superior results in various machine learning applications.

In line with this, several large-scale hashing algorithms have been developed to efficiently estimate or approximate the weighted Jaccard similarity. A series of studies [9, 16, 27, 39, 52] have proposed and refined hashing algorithms based on the rejection sampling technique, which proves to be efficient for dense data vectors. Furthermore, researchers such as Gollapudi and Panigrahy [22], Ioffe [25], Manasse et al. [42] have introduced consistent weighted sampling (CWS), offering a complexity of $O(Kf)$ similar to that of MinHash. CWS operates effectively on relatively sparse data. To improve upon these methods, Li et al. [37] presented Extremal Sampling (ES) based on the extremal stochastic process. Moreover, Li et al. [36] extended the concept of "binning + densification" from OPH to CWS and proposed Bin-wise Consistent Weighted Sampling (BCWS) with a complexity of $O(f)$. BCWS provides a significant

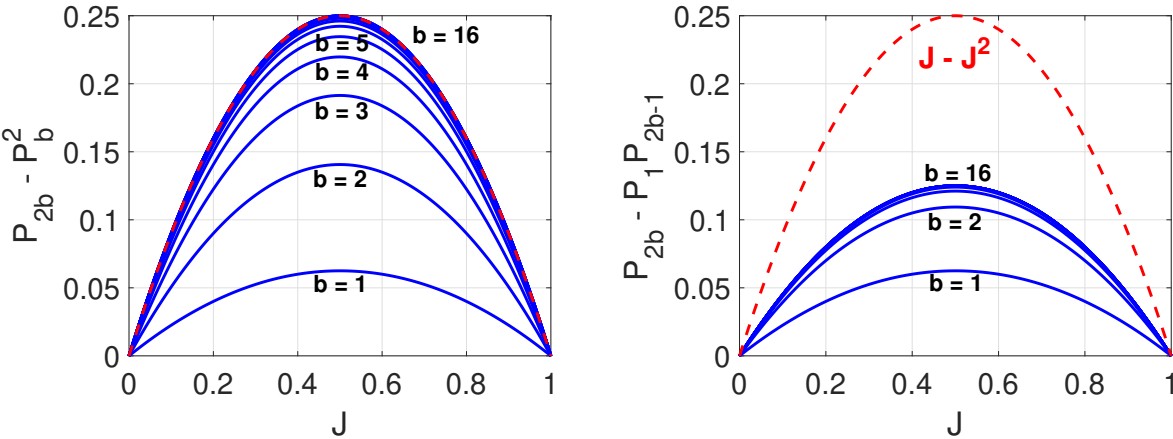

**Figure 2: Plots to verify Lemma 2.2 that $P_{b_1+b_2} - P_{b_1}P_{b_2} \geq 0$. Left panel: $P_{2b} - P_b^2$. Right panel: $P_{2b} - P_1P_{2b-1}$. It is interesting that in both cases, the maximums are attained at $J = 1/2$.**

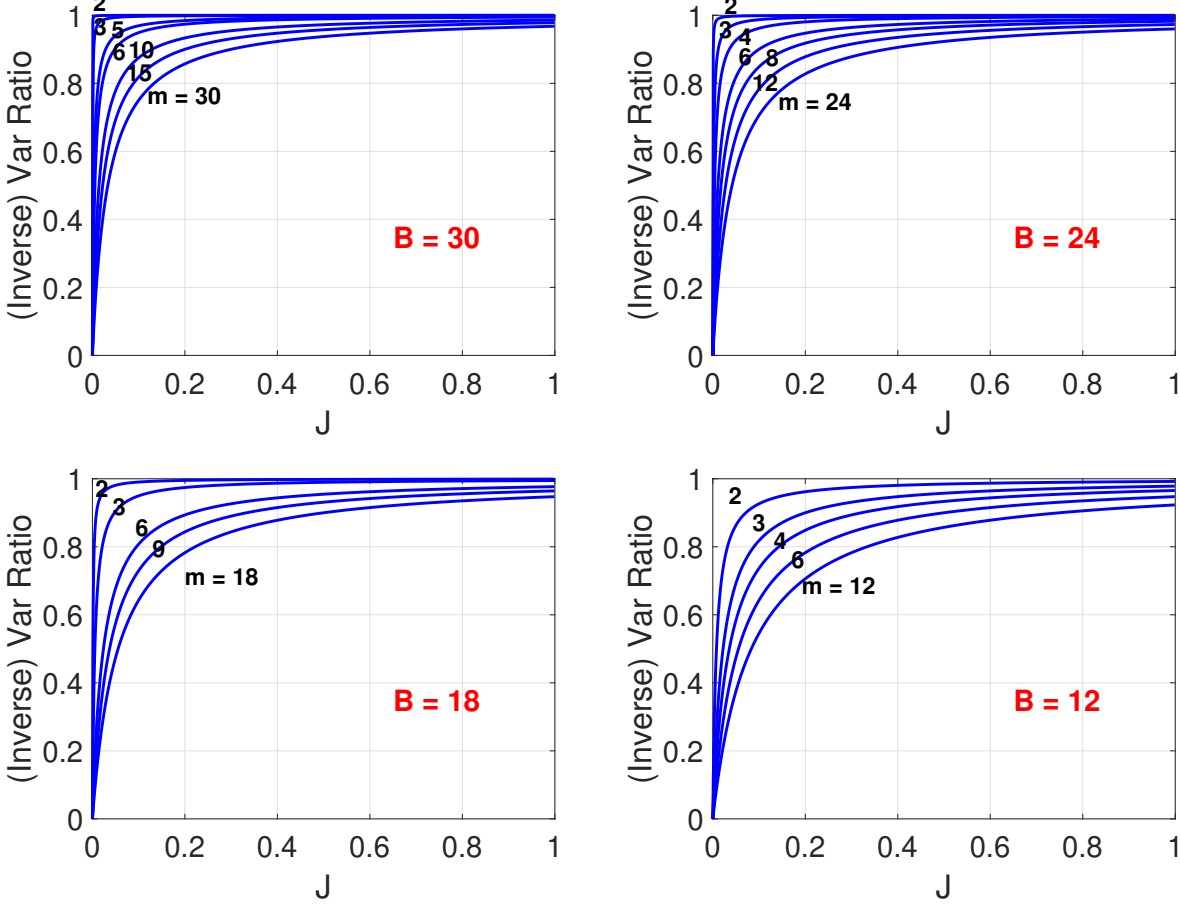

**Figure 3: Plots for $B \in \{30, 24, 18, 12\}$ to illustrate the variance ratio $R_{m,b}$ in (15).**

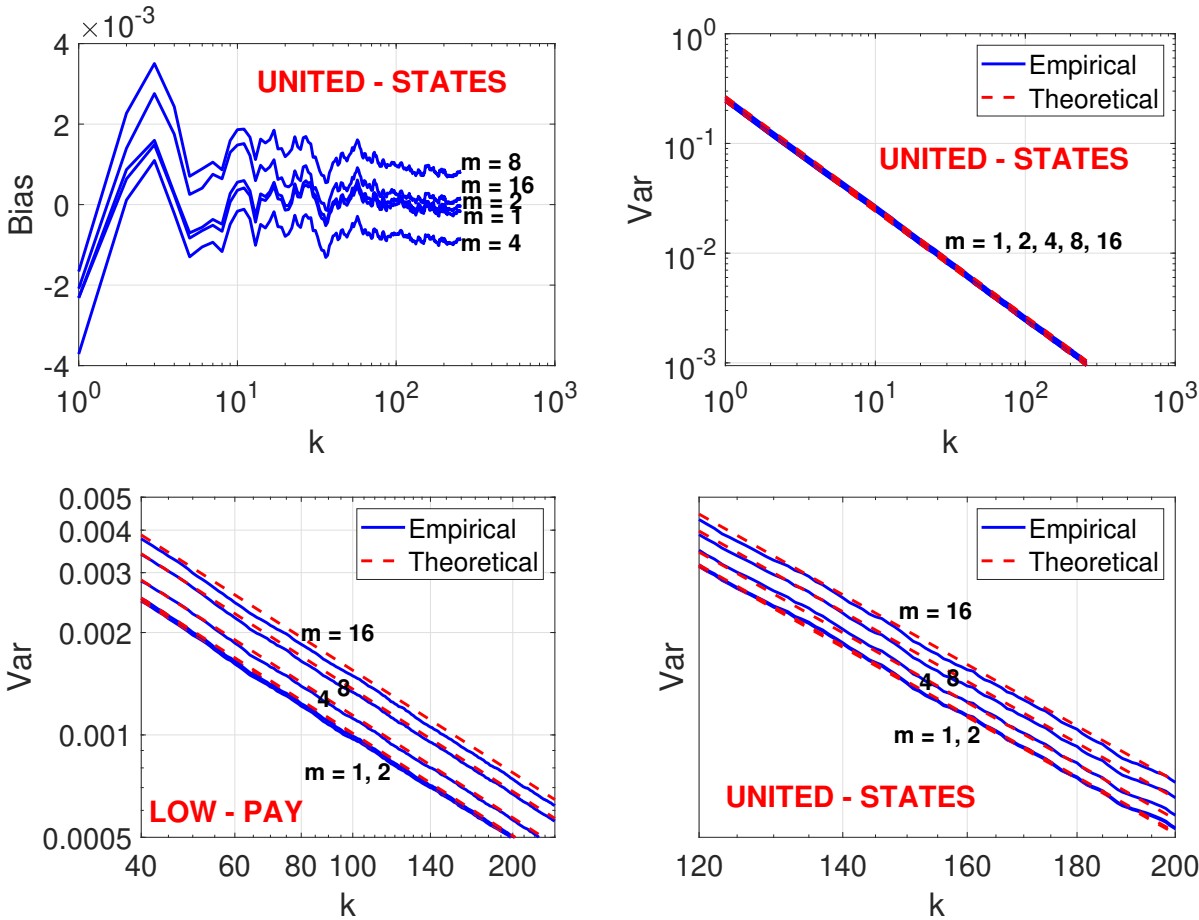

**Figure 4: We use the "Words" dataset [33]. The vector, denoted by "UNITED", stores whether each of $D = 2^{16}$ documents contains the word "UNITED". We use minwise hashing to estimate the Jaccard similarity between the word-pair, e.g., "UNITED–STATES", with $k = 1$ to 1000 hashes. For each hash, we apply Pb-Hash with $m \in \{1, 2, 4, 8, 16\}$. We simulate each case $10^4$ times in order to reliably estimate the biases and variances. The left upper panel plots the biases for each $m$ and $k$. The biases are very small (and the bias$^2$, which will be on the scale as the variance, will be much smaller.). For "UNITED–STATES", the variance curves all overlap in the right upper panel. Thus, we zoom in the plot and present the much magnified portion in the right bottom panel. We can see that, even at such as fine scale, the theoretical variances match the empirical simulations very well. In the left bottom panel, we provide the variance curves on another word-pair "LOW–PAY". Again, the empirical and theoretical curves match quite well. These experiments verify the accuracy of Theorem 2.1, even though it was based on the "Basic Assumption".**

speedup of approximately $K$-fold compared to standard CWS. These advancements in large-scale hashing algorithms facilitate efficient computations and estimations of the weighted Jaccard similarity for diverse datasets. We report the results of Pb-Hash on CWS in Figure 5 and Figure 6.

### 3.3 CWS and Neural Nets

Next we conduct experiments with using CWS hashes for training neural nets. We first break the hash bits into $m$ chunks (for $m = 1, 2, 4, 8$). For each chunk, we connect it with an embedding of size 16. We can simply concatenate all $m$ embeddings, but to reduce the number of parameters and speed up training, we experiment 3 other pooling options: product ("Prod"), mean ("Mean"), and maximum ("Max"). Figure 7 presents the experimental results.

## 4 CONCLUSION

The idea of Pb-Hash, i.e., breaking the bits of one hash value into $m$ chunks, is a very natural one after the work on $b$-bit minwise hashing [34]. At that time, Pb-Hash did not seem to have obvious advantages compared to $b$-bit hashing, because re-generating independent hashes would be always more accurate than re-using the hashes. In recent years, because of the privacy constraint [41], we have started to realize the importance of re-using the hashes. Furthermore, with hashing algorithms used in deep neural nets, the hashed value (i.e., new ID features) is typically connected to an embedding layer and hence there is a strong motivation to break the hash bits into chunks to reduce the embedding size. Also, it is natural to apply Pb-Hash to the original ID features (not the new features obtained via hashing).

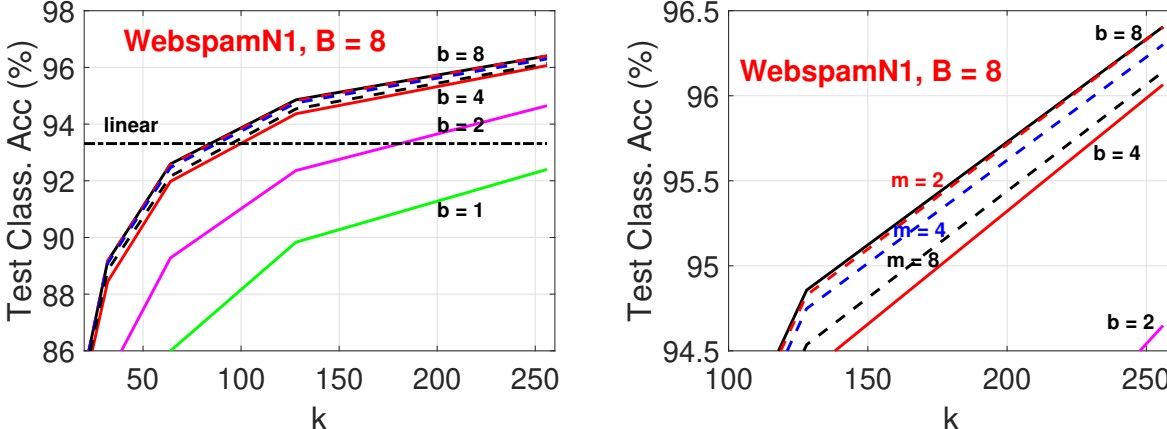

**Figure 5: For the "WebspamN1" dataset (a character 1-gram dataset), we apply CWS and keep $B = 8$ bits for each hash value. We choose $b \in \{1, 2, 4, 8\}$ to run the linear SVM classifier. The left panel shows that when $b = 1$ and $b = 2$, we observe a substantial loss of accuracy. In the right panel, we zoom in to show the Pb-Hash results (i.e., dashed curves). We can see that $m = 2$ and $m = 4$ barely lose any accuracy ($m = 2$ is slightly better than $m = 4$).**

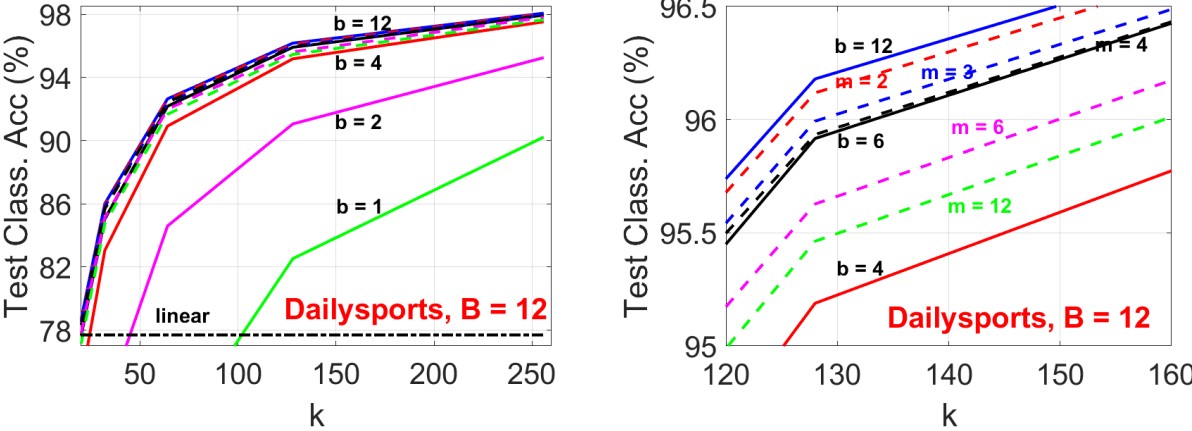

**Figure 6: For the "Dailysports" dataset, we apply CWS and keep $B = 12$ bits for each hash value. We choose $b \in \{1, 2, 3, 4, 6, 12\}$ to run the linear SVM classifier. The left panel shows that when $b = 1$ and $b = 2$, we observe a substantial loss of accuracy. In the right panel, we zoom in to show the Pb-Hash results (i.e., dashed curves). We can see with $m = 2 \sim 4$ the loss of accuracy is small.**

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
