# OpenReview forum: "Pb-Hash: Partitioned b-bit Hashing"
_ACM.org/SIGIR/ICTIR/2024/Conference — ICTIR 2024_

### Official Review · Reviewer_Lp5E · 2024-05-16

**Rating:** -1
**Confidence:** 4

**Objective Part Of Review:**

This paper clearly identifies the challenges of efficiently reusing hashes and proposes the 'partitioned b-bit hashing' (Pb-Hash) method as a solution.

The authors provide clear reasons for reusing hashes and thoroughly introduce the Pb-Hash method, accompanied by theoretical analysis on its impact on accuracy.

The application of Pb-Hash to minwise hashing for binary data and consistent weighted sampling for general non-negative data is well-demonstrated.

However, the paper lacks specific examples or demonstrations of how Pb-Hash could be applied to tasks within Information Retrieval, which could enhance its relevance to the field.

**Subjective Part Of Review:**

While the research questions are interesting, the paper could benefit from explicitly linking its findings to potential applications in Information Retrieval.

Efforts to bridge this gap would increase the appeal of the work to researchers and practitioners in the ICTIR community, as it could potentially offer valuable insights into improving hashing methods for IR tasks.

---

### Official Review · Reviewer_S2bf · 2024-05-16

**Rating:** 2
**Confidence:** 4

**Objective Part Of Review:**

The task is well motivated as efficiency of hashing is important in the large-scale machine learning applications. Using 𝑚 chunks of 𝐵/𝑚 bits may not be as accurate as directly using 𝐵 bits, but the work showed that the accuracy would not drop much for some cases of 𝑚, which indicated viability of the proposed method. The methodology is clearly described. The theoretical analysis appears to be sound while I did not check all the detailed derivations.

**Subjective Part Of Review:**

The proposed partitioned b-bit hashing seems well motivated due to its efficiency and privacy preservation. The proposed method appear novel while it seems inspired by 𝑏-bit minwise hashing. The paper is very relevant to ICTIR as it provides sound theoretical analysis. The effectiveness of Pb-Hash was empirically validated for linear SVM and deep learning.

---

### Official Review · Reviewer_RkTr · 2024-05-17

**Rating:** 1
**Confidence:** 4

**Objective Part Of Review:**

This research investigates a method, called Pb-Hash, to improve the efficiency of neural networks in handling hash codes by dividing them into smaller sections known as chunks. This method can be used either to the original categorical features or to the hashed data. The authors develop a theoretical framework centered around the likelihood of collision between two hash codes. They then put this theory into practice by applying it to Minwise Hashing (MinHash) for binary vectors and Consistent Weighted Sampling (CWS) for general non-negative vectors, evaluating their approach across different datasets. The paper's subject matter fits neatly within the scope of the ICTIR conference. While the paper is generally well-written and includes thorough proofs and experiments on two cutting-edge methods, there are areas where additional clarity could enhance its overall comprehensiveness.
The problem statement in the paper is clearly outlined, focusing on the use of chunks to partition hash code representation to enhance neural network efficiency. The methods employed are adequately described, detailing the theoretical framework and its application to Minwise Hashing (MinHash) and Consistent Weighted Sampling (CWS). However, further elaboration on certain aspects, such as the selection of datasets and the rationale behind them, could enhance clarity.

The results are presented in a clear manner, showcasing the performance of the Pb-Hash method across different datasets and metrics. However, ensuring consistency in the axis ranges for each metric in the plots could improve interpretation.

The paper supports various claims with detailed proofs and experimentation, aligning with the claims made in the abstract and introduction. However, some concepts and notations could benefit from clearer definitions before their usage.

The abstract and introduction provide a general understanding of the paper's objectives and methodology. However, readers may require a deeper understanding of the subject matter to fully grasp the nuances presented.

While no explicit contradictions are evident, providing more detailed explanations or addressing potential limitations could strengthen the paper's argument. For example, a question that might arise from reading the paper concerns whether the performance of the proposed Pb-Hash method is influenced by the choice of data type (such as image, text, or video).

**Subjective Part Of Review:**

The paper is generally easy to read and understand, though certain sections could benefit from clearer explanations or additional details, particularly concerning the choice of datasets and the rationale behind them. Furthermore, ensuring consistency by employing the same range values on both the x-axis and y-axis of figures for each metric could improve readability of the plots, aiding in the interpretation of bias or variance. The problem addressed in the paper, focusing on enhancing neural network efficiency through partitioning hash code representation, is relevant, especially given the increasing importance of optimization techniques in machine learning.

While the methods employed are based on established hashing techniques like Minwise Hashing and Consistent Weighted Sampling, the approach of partitioning hash codes using chunks appears to be an original contribution, offering a novel perspective on improving neural network performance.

---

### Meta-Review · Area_Chair_Z6XX · 2024-05-31

**Recommendation:** Accept (Oral)
**Confidence:** 4

**Metareview:**

This paper presents a method called Pb-Hash for efficient hashing. The problem is well-motivated and the paper includes detailed proofs and experiments to support their claims. The theoretical work of this paper makes it a good fit for ICTIR. The paper would become stronger and receive more attention at the conference by drawing additional connections to IR-specific applications.